# Analgesic effect of diluted nitrous oxide in rehabilitation training of patients with burn scar contracture: Study protocol for a randomized controlled trial

Weifeng Wang[1,2☉], Fei Wang[3☉], Xueling Qiu[2,4], Xiaochen Jiang[1,2], Chenxi Sun[2,5], Fan Sun[2,6], Lu Tang[2]*

1 School of Nursing, Shandong Second Medical University, Weifang, China, 2 Department of Stomatology, The 960th hospital of People's Liberation Army of China (PLA), Jinan, China, 3 Department of Anesthesiology, The 960th hospital of People's Liberation Army of China (PLA), Jinan, China, 4 School of Nursing, Shandong First Medical University, Taian, China, 5 School of Nursing, Jinzhou Medical University, Jinzhou, China, 6 School of nursing, Shandong University of Traditional Chinese Medicine (TCM), Jinan, China

☉ These authors contributed equally to this work.
* tanglu_office@163.com

## Abstract

### Background

Burn patients experience severe pain when undergoing rehabilitation after skin grafting, which negatively affects their recovery. Traditional analgesic methods (such as opioids) have the risk of addictiveness and side effects. Nitrous oxide, which has rapid analgesic and sedative effects, is commonly used for conscious analgesia. The purpose of this study was to determine whether diluted nitrous oxide reduces pain compared to Placebo(oxygen) during rehabilitation after burn surgery.

### Method

This single-center, randomized, double-blind, and controlled trial will enroll 80 patients. Patients ≥ 18 years of age who underwent rehabilitation 1 month after burn surgery with acute pain (VAS ≥ 4) were included. The main exclusion criteria included: pulmonary disease (pulmonary embolism, pneumothorax), intestinal obstruction, etc. Patients were randomly assigned in a 1: 1 ratio to intervention (A) and control (B) groups. Doctors, therapists, patients, and data collectors were unaware of the assignment outcomes. Rehabilitation will be performed by a therapist. The nurse performing the intervention handed the envelope with the patient code and the A or B assignment to the physician. Group A will receive diluted nitrous oxide inhalation plus conventional therapy (without analgesics) 30 minutes once daily for 4 weeks, and Group B will receive oxygen plus conventional therapy (without analgesics) under the same conditions. Assessments will be performed before the intervention (T0), 2

**Data availability statement:** No datasets were generated or analysed during the current study. All relevant data from this study will be made available upon study completion.

**Funding:** 1. Initials of the authors:Wang W.F;Wang F;Qiu X.L;Jiang X.C;Sun C.X;Sun F;Tang L 2. The total amount of funding received for this project is 50,000 yuan. 3. Weifang Medical University 4. https://www.sdsmu.edu.cn/ 5. The sponsor is not involved in any study design, data collection and analysis, decision to publish, or preparation of the manuscript. This study was supported by Weifang Medical University (2023FYM099).The funder was not involved in study design, data collection and analysis, decision to publish, or preparation of the manuscript.

**Competing interests:** The authors have declared that no competing interests exist.

minutes (T1) and 5 minutes (T2) after the start of the intervention, and 5 minutes (T3) after the start of the intervention. The primary outcome was pain score. Secondary outcomes included vital signs, side effects, quality of life score, scar score, need for adjuvant analgesia, therapist and patient satisfaction, and willingness to receive the same gas again.

## Discussion

If the experimental results show that diluted nitrous oxide can bring good analgesic effects without serious side effects, it can improve patients' compliance with rehabilitation treatment and quality of life, and it is even widely implemented in hospitals and rehabilitation institutions.

---

## Introduction

Burn hypertrophic scars(HSs) are a matter of great concern to patients and doctors. Poor wound healing can lead to scar contracture, which is the most harmful complication after common burn surgery [1]. In the process of deep burn wound healing, due to the long-term existence of fibroblasts, the formed scar tissue is easily in a state of poor flexibility and high tension, which will not only aggravate the mechanical stimulation of the burn site, lead to pain and itching, but also cause long-term scar contracture. It will also lead to the loss of motor function of patients and cause bone and joint deformity, thus affecting the quality of life of patients [2–4]. Pressure suits control collagen synthesis by restricting blood supply, oxygen, and nutrient access, thereby inhibiting HSs growth. Although pressure suit therapy has been shown to help reduce HSs, rehabilitation exercises such as exercise, splint, massage, etc. guided by a rehabilitator can effectively prevent or reduce scar contracture and promote functional recovery [5]. Patients with burns may experience decreased mobility caused by fixation sequelae, ventilator dependence, and burn-related catabolic reactions [6]. Hypermetabolism after burns can lead to loss of muscle mass, resulting in rapid onset muscle fatigue [7]. Therefore, the focus of rehabilitation exercise for burn patients is to minimize scar hyperplasia and contracture; Improves flexibility, strength and endurance; And to promote independence in daily activities, including self-care and social and recreational activities.

Operative pain triggered by wound debridement, physical therapy, and rehabilitation exercises exacerbate the suffering of burn patients [8]. Poor pain control can lead to severe sequelae, including post-traumatic stress disorder (PTSD), acute stress disorder (ASD), depression, decreased mobility, delayed return to functional status, opioid dependence, and chronic pain syndrome, among others [9]. Good pain control and wound healing are considered very important throughout the burn recovery process. Despite the significant effect of opioids on acute pain, they carry a high risk of developing adverse effects such as delayed wound healing, respiratory depression, opioid-induced hyperalgesia, pruritus, acute urinary retention, opioid tolerance, dependence, addiction, and overdose-related death [10]. In recent years,

music, hypnosis and virtual reality have been found to be useful non-drug analgesic measures. But because these are just relaxation and distraction techniques, it is difficult to achieve the same level of analgesia as medications.Many studies have shown that nitrous oxide can provide significant analgesic effects in different procedures, for example, dental surgery sedation and analgesia [11], acute injury sedation and analgesia [12], cancer pain [13], bone marrow biopsy [14], invasive operation in pediatrics [15],and labor analgesia,etc [16]. Patients inhale the gas through the self-controlled valve, and the common side effects are dizziness, nausea, vomiting, etc., which can disappear within five minutes after stopping the drug. Under the premise of strict control of contraindications, no serious complications have been found.

There is currently a lack of trials evaluating the efficacy of nitrous oxide in the treatment of pain during rehabilitation training in burn patients. Therefore, the aim of this study was to evaluate the analgesic effect of N2O in burn patients undergoing rehabilitation training by means of a randomized double-blind controlled trial. We hypothesized that its analgesic properties would reduce pain during rehabilitation training in burn patients with few side effects compared to placebo. We designed this study protocol to test and verify this hypothesis.

Objective: This trial was designed to evaluate the efficacy and safety of diluted nitrous oxide in the treatment of acute pain caused by rehabilitation training in burn patients.

## Method

### Design

This study was a single-center, randomized, controlled, double-blind trial that has been registered with the Chinese Clinical Trial Registry (CTR2400089173). The trial protocol was drafted following the checklist of CONSORT statements and SPIRIT Trials guidelines. In this trial, a superiority test will be used and patients will be randomly assigned to two protocol groups at a ratio of 1: 1; Recruitment will begin in April 2025. The schedule of enrolment, and interventions are shown in Fig 1, and the data collection phases of the CONSORT flow chart is shown in Fig 2.

### Ethics and communication

This trial will strictly comply with the Declaration of Helsinki to ensure the rights and safety of participants

1. This study has been approved by the ethics committee (2024−054)

2. All participants will sign the informed consent form in writing after fully understanding the purpose, methods, risks and benefits of the study.

3. All information and data collected during the study will be kept strictly confidential and used exclusively for this study. Data will be anonymized to protect patient privacy.

4. Patients have the right to withdraw from this study unconditionally at any time.

5. The results of the study will be disseminated to the public through relevant academic journals and conferences.

### Participants

All patients with pain caused by rehabilitation training after burn surgery will be invited to participate in this study. In order to reduce complications that may interfere with pain scores during rehabilitation training, we will recruit patients whose burn sites and depths are relatively consistent or close, and the therapists should have more than two years of working experience in the burn department. The inclusion criteria are as follows: men or women at least 18 years old who are willing to participate in the trial and sign the informed consent form; Start functional exercise 3–6 weeks after surgery (with or without skin grafting); Worst pain reported during rehabilitation exercise ≥ 4 (0–10 cm on the visual analog scale (VAS)); There were no postoperative complications. Exclusion criteria included the following: non-primary

| Time point | Study period | | | | | | | |
|---|---|---|---|---|---|---|---|---|
| | Enrolment | Post-allocation | | | | | | Close-out |
| | Patients with burn scar contracture suffer from pain due to rehabilitation training | T0 | T1 | T2 | T3 | T4 | T5 | Data collection measurements |
| **Patients** | | | | | | | | |
| Eligibility screen | √ | | | | | | | |
| Informed consent | √ | | | | | | | |
| Allocation | √ | √ | | | | | | |
| **Interventions** | | | | | | | | |
| Intervention group | | | √ | √ | √ | | | |
| Control group | | | √ | √ | √ | | | |
| **Assessments** | | | | | | | | |
| Pain score | | √ | | √ | √ | | | Visual analogue scale |
| BP | | √ | | √ | √ | | | Non-invasive electronic manometer |
| HR | | √ | | √ | √ | | | Digital monitoring |
| SPO₂ | | √ | | √ | √ | | | Digital monitoring |
| Side effects | | | | | √ | | | Yes/no |
| Scar score | | √ | | | | | √ | Vancouver scale |
| Functional | | √ | | | | | √ | Activity of daily living |
| Satisfaction | | | | | | √ | | Satisfied/dissatisfied |
| Acceptance | | | | | | √ | | Yes/no |
| Adjuvant analgesia | | | | | | √ | | Yes/no |

Abbreviations: BP blood pressure(mm/Hg), HR heart rate(bpm), SPO2 oxygen saturation(%)

**Fig 1. Schedule of inclusion, interventions and assessments.** The leftmost column indicates the contents of recruitment, intervention and evaluation, the checkbox indicates the operation at the corresponding time point, and the rightmost column indicates the evaluation method.

burn patients; Patients with unconsciousness or abnormal mental status that makes it difficult to report pain scores; Patients with residual wounds or active infections in the scar area; Patients with severe inflammation of the treated area; Patients with contraindications to $N_2O$ (epilepsy, pneumothorax, lung cancer, chronic obstructive pulmonary disease and acute respiratory infections, abdominal distension or suspected intestinal obstruction, severe inhalation injury, drug-induced or pathological pulmonary fibrosis, maxillofacial injury, diseases involving the ear, nose, throat, e.g., sinuses, otitis media); Drug dependence or abuse; Pregnant or breastfeeding patients.The main inclusion and exclusion criteria are shown in Table 1.

## Recruitment strategies

The programme will be in the burn department of the 960th Hospital of PLA in China Implement, which is a general hospital in Shandong. Patients who undergo rehabilitation exercises after burn surgery will be the source of recruitment. A member of the study team will be in contact with the case manager, patients and relevant staff in the burn unit for information related to the study. Researchers introduce the details of the study to patients and their attending physicians, including the study purpose, methods, risks, and benefits. Patients will contact the researcher if they are willing to participate in this study. Upon assessment, patients who meet the inclusion criteria will sign informed consent. Diluted nitrous oxide and oxygen will be provided free of charge to patients participating in the study. Medical professionals will provide health and safety protection for subjects. If the study has caused loss or inconvenience to the patient, appropriate compensation is given.

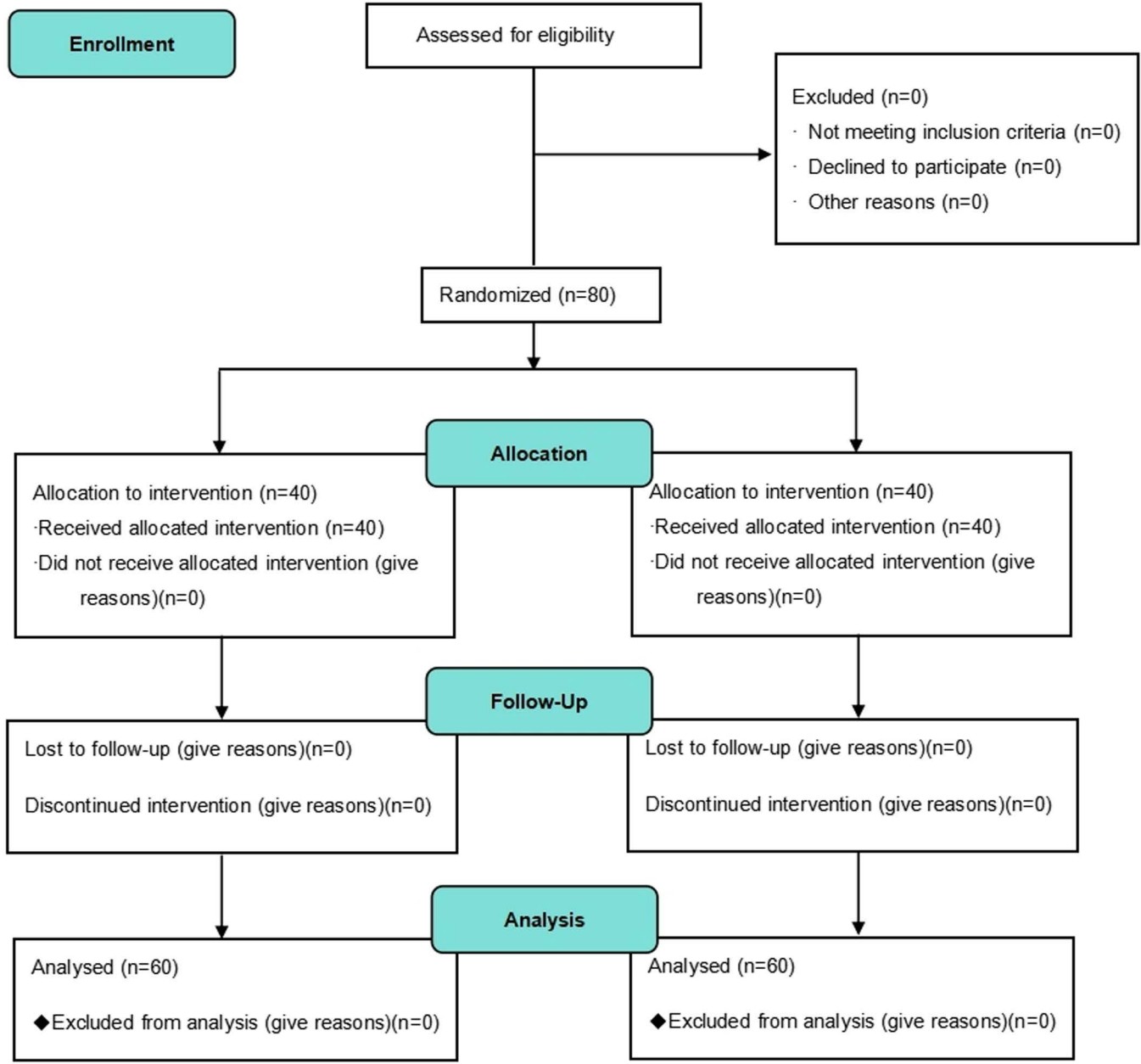

**Fig 2. The CONSORT flow chart.** Patient absence and reasons during recruitment, randomization assignment, assessment and analysis are described.

## Randomization, allocation, concealment and blinding

Randomization will perform by statistical experts, using SPSS 27.0 software randomly assigned patients 1: 1 to the intervention group versus the control group. The experimental group will be given nitrous oxide inhalation and conventional treatment, while the control group will be given placebo (100% oxygen) inhalation and conventional treatment. Before the trial begins, the project manager will prepare a series of opaque envelopes, each containing a card with treatment

**Table 1. Main inclusion and exclusion criteria.**

| Inclusion criteria | Exclusion criteria |
|---|---|
| Burn patients with acute pain caused by functional exercise starting 3–6 weeks after surgery | Patients with psychiatric disorders or altered mental status have difficulty reporting pain |
| Age ≥ 18 years | Patients with non-first burns |
| Worst pain reported during rehabilitation exercise ≥ 4 (0–10 cm on the visual analog scale (VAS)) | Drug dependence or abuse<br>Pregnant or breastfeeding patients |
| There were no postoperative complications | Patients with contraindications to $N_2O$(epilepsy, pneumothorax, lung cancer, chronic obstructive pulmonary disease and acute respiratory infections, abdominal distension or suspected intestinal obstruction, severe inhalation injury, drug-induced or pathological pulmonary fibrosis, maxillofacial injury, diseases involving the ear, nose, throat, e.g., sinuses, otitis media) |

This table lists the inclusion criteria and exclusion criteria. The left side is the inclusion criteria and the right side is the exclusion criteria.

assignment written on it ("nitrous oxide group" or "placebo group"). When the patient meet the inclusion criteria and completed the baseline assessment, the project manager will randomly select an envelope from it and determine the allocation of the patient based on the card inside the envelope. Grouping will be confidential to patients and other investigators. The process of extracting the envelope is free from external interference, and the envelope is immediately destroyed to ensure the concealment of the distribution. The devices for both groups of gases are identical in appearance and are both labeled A or B. Neither the investigators nor the subjects involved in the trial could distinguish gases by smell and appearance. A researcher who is not involved in the evaluation will connect the patient to the gas device during the treatment and ensure the proper inhalation of the gas. Assessments will be performed independently by another investigator according to preset criteria and time points. During this trial, if the patient has serious side effects and other emergencies, the preset emergency unblinding procedure should be followed, and corresponding treatment measures should be taken to ensure the safety of the patient. Statistical analysts are also unaware of patients' grouping and treatment measures.

## Interventions

Rehabilitation training will be carried out by technicians who have worked in the burn department for more than two years, and includes massage and passive movement. When to start rehabilitation exercises and the content depend on the patient's condition. In the early postoperative stage, the skin of patients is relatively tender, and the rehabilitation training that can be given to them is limited. Give priority to passive movement, slightly move the limbs, and try not to touch the newly healed skin, in order to prevent adhesion at the joints. Rehabilitation exercises for patients with good skin healing include active movement and passive movement. When they rely on themselves to move to the maximum angle, the rehabilitation therapist will help them to move to a larger angle, and a lot of pain will occur in this process. Therapists will also massage and release areas where hypertrophic scars are present, with the aim of preventing limited joint movement and scar contracture. Rehabilitation training for burn patients is carried out every day and lasts for a long time. Some patients still need intervention after six to seven years of discharge, which depends on the patient's condition. The patients included in this study were those who needed daily rehabilitation exercises for a short period of time (within one year) after burn.

The intervention group will be given diluted nitrous oxide inhalation on the basis of conventional treatment. Prior to the intervention, a trained nurse will inform the patient of the use and precautions of the inhalation valve. Two minutes before rehabilitation training, patients in the intervention group will inhale premixed 65% N2O through a mask with a

self-controlled inhalation valve until the end of rehabilitation training; The control group will be given continuous oxygen inhalation. These two gases are inhaled and stopped by a patient-controlled valve to facilitate observation of the state of consciousness (the patient will release the valve when the optimal medication status is reached).The intervention will be performed once daily for 30 minutes for 4 weeks.The conventional treatment approach includes the following aspects: 1. The patient uses drugs as directed by the doctor during hospitalization. 2. Therapist's psychological counseling to patients before and during operation. Patients in both the intervention and control groups will have continuous blood oxygen saturation and heart rate monitored, and blood pressure will be measured before, during and after the intervention. If an adverse reaction occurs, the inhalation should be stopped immediately and oxygen should be given. The adverse reaction will disappear within a few minutes.

## Outcome measures

**Baseline data.**  Demographic data (gender, age, height, weight, country, occupation, etc.) and clinical characteristics (burn site and depth, current pain level, Vancouver scar score) will be recorded for all subjects at T0 (two minutes before intervention). Vital signs, such as blood pressure (BP), heart rate (HR), blood oxygen saturation (SPO2), were also recorded at T0.

**Primary outcome.**  The primary outcome was the the degree of pain reduction in patients. Patients' pain levels will be assessed using the VAS scale at T0 (before intervention), T2 (two minutes after the start of the intervention) and T3 (five minutes after the start of the intervention). VAS allowed subjects to select a point on a straight line 10 cm long (0: painless, 10: worst possible pain) to report their own pain level [17]. The VAS scale is easy to understand and implement, and it is a widely used pain assessment scale, and it does not interrupt the intervention process of subjects. Gas suction starts at T1 and stops at T4.

**Secondary outcome.**  The patient's vital signs, including: oxygen saturation, heart rate, blood pressure, will be measured at T0 (before inhalation), T2 (two minutes after inhalation), and T3 (five minutes after inhalation). At the end of the procedure, the subjects will be asked what side effects occurred during the intervention and the consumption of adjunctive analgesics will be recorded. Patient and technician satisfaction and willingness to re-use analgesic gas were also secondary outcomes. Vancouver scar assessment scale and ADL scale were used to observe the scar improvement and functional rehabilitation of patients at T5(Four weeks after intervention), respectively. The purpose of this study is to control acute pain during burn rehabilitation, and the effect of scar treatment is indirect, so scar score and functional improvement are set as secondary outcomes.

## Sample size estimation

The degree of pain reduction was the primary outcome of this study. Data for sample size calculations are based on the change in pain levels at T3, as this is the peak of nitrous oxide effect.Percentage changes in VAS scores ≥ 30% were considered clinically meaningful in settings with large differences in basal pain levels. Based on a set of VAS data we previously collected, the effect size at this time was calculated to be 1.68. We employed PASS 17.0 software to calculate the sample size.To assess the effectiveness of this gas analgesic effect, for the bilateral test, we set α (class I error) to 0.05 and test efficacy 1-β to 90% (β = 0.1).After our calculations, using a two-sided, two-sample equal variance t-test, the ability to reject the null hypothesis of zero effect size with a group sample size of 18(9 cases per group) was 91.67% when the overall effect size was 1.68 and the level of significance (alpha) was 0.05.In order to meet the standards of the State Food and Drug Administration and reduce the inadequacy of this study, we decided to recruit a total of 80 patients(40 cases per group).

## Data management and analysis

In order to ensure the accurate entry of data, two independent personnel enter the data respectively and compare them to ensure consistency. In order to improve quality, researchers are trained before studying. The senior academic statisticians

in the group regularly checked the data, including missing values, outliers, etc., and corrected them in time.Missing data from subjects who withdraw from the study will be imputed using multiple imputation methods. Shortly after project initiation, a data monitoring committee (DMC) composition and audit system will be established to monitor data collection and management and patient safety throughout the study. Four professional therapists and nurses, two pain management specialists, and a senior academic statistician who serves as chair of the board will serve as members. Patient safety is monitored and managed by professional therapists and pain management experts.All information will be stored confidentially in the office. Replace all identifying information of the participant, such as name and gender, with a code and always keep it confidential, with only the project manager or attending physician knowing the patient's personal information. In addition, none of the reports about this study will disclose the personal information of patients.

Statistical analysis will perform using spss.27, with an alpha set at 0.05. The mean (standard deviation), median (interquartile range), minimum value, maximum value and proportion (95% confidence interval) of quantitative data will be listed, and the count data will be described in quantity and percentage. Mean comparison between the two groups will be performed using t-test (normal distribution) and Mann-Whitney test (non-normal distribution). Chi-square test will be used to compare proportions.A value of $P < 0.05$ will be considered statistically significant.

## Safety

Side effects of nitrous oxide include dizziness, headache, nausea, vomiting, drowsiness, euphoria, lack of oxygen, arterial hypotension, bradycardia, etc. Therapists monitor and record in a timely manner during administration and after discontinuation. When the above adverse reactions occur, stop gas inhalation and give oxygen to the patient, then the discomfort will be relieved within five minutes [18–19].

## Trial status

The trial is registered in the Clinical Trials Registry: ChiCTR2400089173. Participant recruitment began on April 7, 2025 and is expected to be completed within eight months.Data collection and analysis expected to be completed by February 2026. At the time of writing, participating clinical centers are preparing for active recruitment.We expect 10 eligible patients to join the study each month.

## Discussion

Studies have shown that the incidence of scar contracture at hospital discharge is high, about 40% to 55%, and therefore requires optimal ongoing rehabilitation and care. Patients who healed within 14–21 days had a 30% incidence of scar contracture, compared with 78% in patients over 21 days [20]. Scar contracture can affect people's ability to participate in daily life. Neuropathic pain, pruritus, itching, and stiffness can lead to impaired function in burn sufferers. Daily scar massage prevents and improves scar contracture by improving scar-related pain, itching, flexibility and thickness [21]. Patients should receive scar management techniques as soon as the burn wound is closed and there is no risk of skin rupture or abrasion.

Recovery from burns begins within minutes of admission and can last months or even years. This long-term phase of rehabilitation may include reconstructive surgery and lifelong services to treat contractures exacerbated by patient growth and aging.Pain is a challenge faced by burn patients in rehabilitation. Proper pain management can increase patient compliance and satisfaction, and also reduce the difficulty and stress of therapists' work.

Our initial interviews found that pain control in burn patients relies primarily on opioids. However, many patients and therapists have shown resistance to its application due to concerns about its side effects. In fact, it also involves cultural differences between different countries and regions. In recent years, European and American countries have strengthened the management of opioid analgesics to resist drug abuse, while East Asia has always been wary of this drug [22]. Patients and therapists are worried that frequent use of opioids will increase the risk of side effects, and at the same time, they both show a desire to relieve pain appropriately.

Nitrous oxide has shown many advantages in conscious analgesia compared to conventional opioid analgesics. It is an inhaled gas by self-administration and has analgesic, sedative, anxiolytic characteristics. Nitrous oxide has been widely used in recent years, such as dentistry, emergency department, obstetrics and gynecology, pediatrics, postoperative analgesia and other clinical disciplines. It has been reported that a mixture of 20% nitrous oxide and 80% oxygen has an equivalent to 15 mg Comparable analgesic effect of morphine as well as good sedation [23]. It begins to work 30 seconds after inhalation, peaks in two minutes, and the effect disappears within 5 minutes after withdrawal, with mild adverse effects.In addition, it can produce analgesic effects without changing consciousness and cognitive state, which can improve the cooperation of burn patients.

Although nitrous oxide is simple to operate and has mild side effects, it can also be life-threatening in some cases. The replacement of oxygen in the alveoli by nitrous oxide leads to a sudden drop in the partial pressure of blood oxygen, which may lead to hypoxia and asphyxia, which often occurs when the oxygen supply equipment fails and excessive sedation leads to spontaneous respiratory depression. Therefore, it is recommended to use a pre-mixed oxygen/nitrous oxide mixture and monitor with a pulse oximeter. Craniocerebral patients, intestinal obstruction and other patients will have fatal consequences such as cerebral hernia and intestinal perforation. Safety management personnel must strictly grasp the contraindications of nitrous oxide, learn to observe the symptoms of side effects and master first aid measures during training. Diluted nitrous oxide is often stored in cylinders and used through valves to facilitate transportation. Nitrous oxide gas is not expensive, and cylinders and valves can be used repeatedly. Therefore, regular maintenance of equipment and use according to specifications can greatly save costs. Monitoring patients' vital signs outside surgery is a tricky task. Generally speaking, we can only monitor through portable sphygmomanometers and finger clip oximeters. The disadvantage is that we cannot monitor blood pressure dynamically, but this is enough for the key indicators that need to be monitored.

Therefore, we designed this study to explore the effects of nitrous oxide on pain relief during rehabilitation training in burn patients. If feasible, the widespread application of nitrous oxide can greatly alleviate the pain of burn patients.

## Limitation

There are indeed some limitations in this study. First of all, the pain score is a subjective result and is susceptible to the subjective influence of patients. Vital signs are objective indicators, but the degree to which they reflect pain varies from person to person. Secondly, there are differences in outcome measures for patients due to differences in the site, degree, and depth of their burns, so we will include patients with similar conditions as much as possible, but cannot completely eliminate this difference.Finally, pain perception is correlated with emotion. Oxygen may produce slight relaxation or euphoria, and even if it does not directly relieve pain, it may indirectly affect the patient's subjective feelings. Pure oxygen as a placebo may not be a completely inert control, and its effect may lead to an overestimation of the placebo effect and an underestimation of the true analgesic effect of nitrous oxide.

## Conclusion

To the best of our knowledge, this study is the first randomized controlled trial to evaluate the effectiveness of N2O in the treatment of acute pain in patients undergoing rehabilitation after burns. If this treatment appears to be beneficial for burn patients undergoing rehabilitation, this study could help generate preliminary guidelines for pain management. We will disseminate the results of this research in international journals and conferences.

## Supporting information

**S1 File. This is the SPIRIT 2013 checklist containing the items recommended to address in a clinical trial protocol and related documents, marking where each item is located in the text.**
(DOC)

**S2 File. This is the material submitted when applying for ethics, which contains a brief description of the study protocol and informed consent form.**
(DOCX)

## Acknowledgments

We sincerely thank all those involved in the study and all investigators and clinical staff involved in the trial for their efforts.

## Author contributions

**Conceptualization:** Weifeng Wang, Chenxi Sun.

**Methodology:** Fei Wang, Xueling Qiu, Xiaochen Jiang.

**Project administration:** Fei Wang, Lu Tang.

**Software:** Weifeng Wang, Fan Sun.

**Supervision:** Lu Tang.

**Writing – original draft:** Weifeng Wang.

**Writing – review & editing:** Weifeng Wang.

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
