## [Decision Letter · Decision Letter 0]

7 Jul 2025

Dear Dr. Tang,

Thank you for submitting your manuscript to PLOS ONE. After careful consideration, we feel that it has merit but does not fully meet PLOS ONE’s publication criteria as it currently stands. Therefore, we invite you to submit a revised version of the manuscript that addresses the points raised during the review process.

We look forward to receiving your revised manuscript.

Kind regards,

Jonathan Bayuo, PhD

Academic Editor

PLOS ONE

3. Please include a caption for figure 1 and 2.

Additional Editor Comments:

Thanks to the authors for their submission. In addition to the comments raised by the reviewers, please see the following comments for your kind consideration:

1. It is unclear how the authors will identify "All patients with pain caused by rehabilitation training after burn surgery"? This remains unclear and warrants further clarification.

2. Is the study targeting in-patients or burn survivors in the post-discharge period?

3. For adult burn survivors, i wonder if the visual analogue scale remains more appropriate than other objective pain assessment tools such as the numerical rating scale?

4. Please provide details of the conventional treatment approach.

Reviewers' comments:

Reviewer's Responses to Questions

**Comments to the Author**

1. Does the manuscript provide a valid rationale for the proposed study, with clearly identified and justified research questions?

Reviewer #1: Yes

Reviewer #2: Yes

2. Is the protocol technically sound and planned in a manner that will lead to a meaningful outcome and allow testing the stated hypotheses?

Reviewer #1: Partly

Reviewer #2: Partly

3. Is the methodology feasible and described in sufficient detail to allow the work to be replicable?

Reviewer #1: Yes

Reviewer #2: Yes

4. Have the authors described where all data underlying the findings will be made available when the study is complete?

Reviewer #1: No

Reviewer #2: Yes

5. Is the manuscript presented in an intelligible fashion and written in standard English?

Reviewer #1: Yes

Reviewer #2: Yes

You may also provide optional suggestions and comments to authors that they might find helpful in planning their study.

Reviewer #1: This protocol outlines a well-designed, clinically relevant RCT addressing an important pain management challenge in burn rehabilitation. Its strengths lie in its focus, the choice of intervention, and the rigorous double-blind, placebo-controlled design.

However, critical methodological details are currently missing, most notably the justification for the sample size. The protocol states will enroll 120 patients, but provides no justification or power calculation. This is a major omission. Why 120? Without this, the study risks being underpowered to detect a meaningful difference.

Clarification on rehab standardization, specific outcome measurement tools also essential. While therapists perform rehab, the protocol lacks detail on the standardization of the rehabilitation exercises and stretches across patients and sessions. Variability in the rehab stimulus could be a significant confounder for pain scores. More accurate tools are necessary for measurement and evaluation.

Addressing these points would significantly strengthen the protocol's validity and the potential impact of the study findings.

The discussion point about widespread implementation is optimistic but glosses over practical barriers like equipment cost, staff training requirements for safe N₂O administration, and monitoring infrastructure needed, especially outside operating rooms.

Reviewer #2: Thank you for submitting this important and timely study protocol. Your proposed randomized controlled trial addresses a critical and often overlooked challenge in burn rehabilitation—pain management during physical therapy. The study is well-conceived, with a clear scientific rationale supported by current literature, and has the potential to make a valuable contribution to improving compliance and patient outcomes in burn care.

The protocol demonstrates commendable strengths, including:

A well-articulated clinical rationale and literature background.

Adherence to CONSORT and SPIRIT guidelines.

Clearly defined inclusion and exclusion criteria.

Strong ethical safeguards, including detailed consent and safety monitoring procedures.

Well-designed randomization and blinding protocols that minimize bias.

That said, I would appreciate clarification on a few important points:

1. Sample Size Discrepancy:

There appears to be an inconsistency in the reported sample size. The summary section on first page mentions a total of 80 participants, whereas another part of the manuscript references 120 participants. Please clarify the actual planned sample size and ensure consistency throughout the manuscript.

2. Duration and Frequency of Intervention:

The manuscript does not clearly state how long participants will receive rehabilitation training (in terms of weeks or months), nor how frequently sessions will occur. Providing this information is important for assessing the feasibility and clinical relevance of the study. Please clarify the duration of the intervention and the frequency of treatment sessions.

3. Clarity of Study Timeline and Terminology:

The use of T1 to T5 to describe time points at is somewhat confusing without adequate context. What time point is T5?

As T1 to T5 indicates time point at which measurements are taken on a particular visit, in cases of multiple visits there should be T1 to T5 for each study visit.

4. 100 % oxygen

While using 100% oxygen as the control is understandable, it may not be entirely inert. Oxygen can have mild psychoactive effects, and this could potentially confound pain perception. I recommend briefly discussing this in the limitations section to acknowledge any placebo-related influence on outcomes.

5. Data Transparency and Accessibility:

The ethical considerations section is comprehensive. For greater transparency and reproducibility, I recommend specifying where the final dataset will be stored following completion of the study—e.g., an institutional repository, open-access platform, or other publicly accessible database—particularly if the results are intended for publication.

I appreciate your thoughtful work on this important topic.

**Do you want your identity to be public for this peer review?** For information about this choice, including consent withdrawal, please see our Privacy Policy

Reviewer #1: No

Reviewer #2: No

---

## [Author Response · Author response to Decision Letter 1]

15 Jul 2025

To editor:

Thank you very much for helping us point out the problem, here is my response item by item.If you still have any questions, please feel free to contact me. I wish you the best!

1 The manuscript has been revised to meet the style requirements of PLOS ONE

2 I have removed the ethical statement from the manuscript except the method section.

3 I have added caption at the reference figure

4 Shandong Second Medical University The previous name of the university was Weifang Medical University. After this project was approved, the school was renamed Shandong Second Medical University. I have changed the funding information in the manuscript to Weifang Medical University. If you still need to modify Please let me know.

Comments from other editors:

1 In the method-Participants, "All patients with pain caused by rehabilitation training after burn surgery will be invited to participate in this study." We then described the inclusion criteria: "Worst pain reported during rehabilitation exercise ≥ 4 (0-10 cm on the visual analog scale (VAS))." Different patients have differences in their condition and pain degree. The main purpose of this study is to reduce acute pain during rehabilitation training in burn patients, so we took pain degree as an inclusion criterion. This inclusion criterion explains this sentence.

2 The rehabilitation of burn patients needs professional rehabilitation trainers to complete it. The subjects of this study are hospitalized patients.

3 Visual analog scale has the advantages of accuracy, simplicity and high sensitivity, and is widely used in clinical and scientific research work. Another advantage of VAS is that its value changes continuously and can be used for parameter verification. The use of digital rating scale requires patients to have certain abstract scale comprehension ability and word reading comprehension ability. Therefore, in order to avoid subsequent special situations, we chose to use visual analog scales.

4 conventional treatment approach I have added to the interventions section of the manuscript.

Reviewer 1:

The question you asked is very important, thank you so much! We have made changes

1. We have described the calculation of sample size in more detail in the Sample size estimation section of the article. The calculation results show that when the sample size of the group is 18 (9 cases in each group), it can meet the requirements of the parameters in this paper. However, as requested by the State Food and Drug Administration, to find out if the gas had more side effects and to increase the credibility of the study, we expanded the sample size to 120 (60 cases per group)

2. I have added details about rehabilitation exercises to the intervention section. In terms of outcome indicators, some editors also raised this issue. We used the visual analog scale (VAS) to assess the degree of pain because it has the advantages of accuracy, simplicity and high sensitivity. In fact, there are more accurate measurement tools such as biomarkers, but because we have many measurement nodes, multiple blood sampling in a rehabilitation training may arouse patient resentment, and the inflammatory response of burn patients may affect its accuracy. Therefore, we currently believe that VAS is an economical, simple and accurate measurement tool suitable for this test. We will also monitor the changes of patients' blood pressure and heart rate to assist the test.

3. We do ignore this point. The problems actually faced by nitrous oxide applications have been described in the discussion section.

Reviewer 2:

The problems you pointed out can make up for our great shortcomings. Thank you very much for your careful review. We have made corrections.

1. This has been corrected in the article. The calculation of sample size is described in more detail

2. Details about rehabilitation exercises have been added to the intervention section.

3. T1 is a time point at which gas inhalation is started, and T4 is a time point at which gas inhalation is stopped. The onset time of nitrous oxide is mentioned in our discussion section from 30 seconds to two minutes, and the settings of T2(two minutes after inhalation) and T3 (five minutes after inhalation) are determined according to this pharmacological property. There is a real problem with the description of T5, which we have revised to four weeks after intervention. In Fig.1, it is clear at which time point each outcome measure was evaluated.

4. We did ignore this point, and we have added a relevant description in the limitations section.

5. We added this to the Data Availability section.

---

## [Decision Letter · Decision Letter 1]

4 Aug 2025

Dear Dr. Tang,

Thank you for submitting your manuscript to PLOS ONE. After careful consideration, we feel that it has merit but does not fully meet PLOS ONE’s publication criteria as it currently stands. Therefore, we invite you to submit a revised version of the manuscript that addresses the points raised during the review process.

We look forward to receiving your revised manuscript.

Kind regards,

JONATHAN BAYUO, PhD

Academic Editor

PLOS ONE

Journal Requirements:

Reviewers' comments:

Reviewer's Responses to Questions

**Comments to the Author**

1. Does the manuscript provide a valid rationale for the proposed study, with clearly identified and justified research questions?

Reviewer #1: Yes

Reviewer #2: Yes

2. Is the protocol technically sound and planned in a manner that will lead to a meaningful outcome and allow testing the stated hypotheses?

Reviewer #1: Yes

Reviewer #2: Yes

3. Is the methodology feasible and described in sufficient detail to allow the work to be replicable?

Reviewer #1: Yes

Reviewer #2: Yes

4. Have the authors described where all data underlying the findings will be made available when the study is complete?

Reviewer #1: Yes

Reviewer #2: Yes

5. Is the manuscript presented in an intelligible fashion and written in standard English?

Reviewer #1: Yes

Reviewer #2: Yes

You may also provide optional suggestions and comments to authors that they might find helpful in planning their study.

Reviewer #1: The author has responded to all comments and explained the specific nature of certain data that could not be provided in accordance with the study protocol. The authors conclude that "this study is the first randomized controlled trial to evaluate the effectiveness of N2O in treating acute pain in patients undergoing rehabilitation after burns”. Should this treatment prove beneficial for burn patients in rehabilitation, the findings could inform the development of preliminary guidelines for pain management. This study addresses a novel research question and holds significant clinical applicability. The results are highly anticipated and warrant further investigation.

Reviewer #2: Thank you for your detailed revisions. Most of my concerns have been addressed satisfactorily. However, one point remains unclear regarding the frequency of the intervention. The duration is clear which is 30minutes.

In the abstract, it states:

"Group A will receive diluted nitrous oxide inhalation plus conventional therapy (without analgesics) once daily for 30 minutes, and Group B will receive oxygen plus conventional therapy (without analgesics) under the same conditions."

While this describes the daily frequency and session length, it does not clarify the total duration of the intervention. Is this a one-time session, a daily intervention for a set number of days, or several sessions per week over multiple weeks?

You mention that T5 is the final follow-up point after 4 weeks, suggesting a 4-week intervention period. If this is the case, I recommend that the full duration of the rehabilitation (e.g., “once daily for 30 minutes over 4 weeks”) be clearly specified in both the abstract and the methods section to ensure reproducibility and clarity for future researchers.

Thank you again for your excellent work.

**Do you want your identity to be public for this peer review?** For information about this choice, including consent withdrawal, please see our Privacy Policy

Reviewer #1: No

Reviewer #2: No

---

## [Author Response · Author response to Decision Letter 2]

7 Aug 2025

To editor:

We don't seem to find that you have raised a new question. Thank you very much for your help! Please feel free to contact me if you still have questions. I wish you all the best!

Reviewer 1:

Thank you so much for your recognition and help with our manuscript!

Reviewer 2:

What you said is really important, and we have corrected it in the text. Thank you so much for your questions!

---

## [Decision Letter · Decision Letter 2]

24 Aug 2025

Dear Dr. Tang,

Thank you for submitting your manuscript to PLOS ONE. After careful consideration, we feel that it has merit but does not fully meet PLOS ONE’s publication criteria as it currently stands. Therefore, we invite you to submit a revised version of the manuscript that addresses the points raised during the review process.

We look forward to receiving your revised manuscript.

Kind regards,

JONATHAN BAYUO, PhD

Academic Editor

PLOS ONE

Journal Requirements:

Reviewers' comments:

Reviewer's Responses to Questions

**Comments to the Author**

1. Does the manuscript provide a valid rationale for the proposed study, with clearly identified and justified research questions?

Reviewer #3: Partly

2. Is the protocol technically sound and planned in a manner that will lead to a meaningful outcome and allow testing the stated hypotheses?

Reviewer #3: Partly

3. Is the methodology feasible and described in sufficient detail to allow the work to be replicable?

Reviewer #3: Yes

4. Have the authors described where all data underlying the findings will be made available when the study is complete?

Reviewer #3: No

5. Is the manuscript presented in an intelligible fashion and written in standard English?

Reviewer #3: Yes

You may also provide optional suggestions and comments to authors that they might find helpful in planning their study.

Reviewer #3: The labels for time points of measuring pain are inconsistent. E.g. on page 2, it indicates that assessments will be performed before the intervention (T0), 2 minutes (T1) and 5 minutes (T2) after the start of the intervention, and 5 minutes (T3) after the end of the intervention. While on page 10 it states that patients' pain levels will be assessed using the VAS scale at T0 (before inhalation), T2 (two minutes after inhalation) and T3 (five minutes after inhalation). Gas suction starts at T1 and stops at T4. Please revise them and make it consistent and clear.

On page 13, the primary outcome is mentioned as the degree of pain. But for the sample size estimation on page 14, the primary outcome is the degree of pain reduction. Please make them consistent. It is reasonable to use the degree of pain reduction as the primary outcome, but it needs to be clearly described at which time point after intervention the degree of pain reduction will be examined. Please provide the VAS data previously collected for each pilot group to show evidence of observed effect size at which time point. This effect size should have been the difference of the degree of pain reduction between two groups related to the standard deviation of the degree of pain reduction. If the estimated sample size is 9 case/group, the recruitment of 60 case/group may be overpowered and will take long time to recruit patients. This increased sample size is not necessary to detect the difference of the degree of pain reduction according to the assumed effect size, unless you plan to estimate more reliable 95% confidence interval for average degree of pain reduction in each group separately, and/or run regression models to evaluate factors that would affect the degree of pain reduction. It needs more descriptions in statistics considerations and data analysis for such additional plans.

Page 6, it says that recruitment will begin in December 2024. But on page 17, it says that participant recruitment began on April 7, 2025. How many eligible patients are expected to join the study each month within 8 months recruitment?

**Do you want your identity to be public for this peer review?** For information about this choice, including consent withdrawal, please see our Privacy Policy

Reviewer #3: No

---

## [Author Response · Author response to Decision Letter 3]

25 Aug 2025

To editor:

We don't seem to find that you have raised a new question. Thank you very much for your help! Please feel free to contact me if you still have questions. I wish you all the best!

Reviewer 3:

Thank you very much for your careful review, the questions you have raised are very important.

1. In the description of the measurement time point, changing "inhalation" to "intervention" is indeed clearer and more consistent, and we have modified it in the text.

2. There are indeed inconsistencies in the description of pain in the sample size estimation, which we have revised to the degree of pain reduction.

3. A clear description of the intervention time points we added to the sample size estimation section.

4. We have described the calculation of sample size in more detail in the Sample size estimation section of the article. The calculation results show that when the sample size of the group is 18 (9 cases in each group), it can meet the requirements of the parameters in this paper. This gas reported a low incidence of side effects and adverse reactions, as requested by the State Food and Drug Administration, to find out if the gas had more side effects, We expanded the sample size.A larger sample size can add credibility to the study, but the 120 sample size does present some difficulties during the collection process. Therefore, following your suggestion, we finally revised the sample size to 80 cases (40 cases per group).We expect 10 eligible patients to join the study each month.

5.Due to the purchase of gas and equipment, we delayed the recruitment time, and the article was not revised in time. Corrections have been made now.

6.The T0 and T3 time point data for the preliminary experiment were as follows (A: intervention group B: placebo group).This is only used to calculate sample size.

VAS 1 2 3 4 5 6 7 8 9 10

A(T0) 8 9 8 6 7 8 5 6 8 6

A(T3) 6 6 5 4 5.5 5 4.5 5 4 3

Δ -2 -3 -3 -2 -1.5 -3 -0.5 -1 -4 -3

B(T0) 8 7 6 7 5.5 7 6 6 7 7

B(T3) 6 6 5 6 5 6.5 5 5 5.5 6

Δ -2 -1 -1 -1 -0.5 -0.5 -1 -1 -1.5 -1

It has been proposed in this paper that the sample size is calculated using PASS 15 software.

σ = √[(σ₁² + σ₂²)/2] d = (μ1 - μ2) / σ

Alternative Hypothesis= Two-Sided

Power= 0.9

Alpha= 0.05

Any help is greatly appreciated! I wish you all the best!

---

## [Decision Letter · Decision Letter 3]

5 Sep 2025

Dear Dr. Tang,

Thank you for submitting your manuscript to PLOS ONE. After careful consideration, we feel that it has merit but does not fully meet PLOS ONE’s publication criteria as it currently stands. Therefore, we invite you to submit a revised version of the manuscript that addresses the points raised during the review process.

We look forward to receiving your revised manuscript.

Kind regards,

JONATHAN BAYUO, PhD

Academic Editor

PLOS ONE

Journal Requirements:

Reviewers' comments:

Reviewer's Responses to Questions

**Comments to the Author**

1. Does the manuscript provide a valid rationale for the proposed study, with clearly identified and justified research questions?

Reviewer #3: Yes

2. Is the protocol technically sound and planned in a manner that will lead to a meaningful outcome and allow testing the stated hypotheses?

Reviewer #3: Yes

3. Is the methodology feasible and described in sufficient detail to allow the work to be replicable?

Reviewer #3: Yes

4. Have the authors described where all data underlying the findings will be made available when the study is complete?

Reviewer #3: No

5. Is the manuscript presented in an intelligible fashion and written in standard English?

Reviewer #3: Yes

You may also provide optional suggestions and comments to authors that they might find helpful in planning their study.

Reviewer #3: In the primary outcome, T3 indicates five minutes after the start of the intervention. But in the abstract, T3 is after the end of the intervention. Please make sure the labels of time points are consistent through the text.

**Do you want your identity to be public for this peer review?** For information about this choice, including consent withdrawal, please see our Privacy Policy

Reviewer #3: No

---

## [Author Response · Author response to Decision Letter 4]

5 Sep 2025

To editor:

We don't seem to find that you have raised a new question. Thank you very much for your help! Please feel free to contact me if you still have questions. I wish you all the best!

Reviewer 3:

Thank you very much for your careful review!We have made changes in the abstract.

I wish you all the best!

---

## [Decision Letter · Decision Letter 4]

14 Sep 2025

Analgesic effect of diluted nitrous oxide in rehabilitation training of patients with burn scar contracture: study protocol for a randomized controlled trial

PONE-D-25-25104R4

Dear Dr. Tang,

We’re pleased to inform you that your manuscript has been judged scientifically suitable for publication and will be formally accepted for publication once it meets all outstanding technical requirements.

Kind regards,

JONATHAN BAYUO, PhD

Academic Editor

PLOS ONE

Additional Editor Comments (optional):

Reviewer #3:

Reviewers' comments:

Reviewer's Responses to Questions

**Comments to the Author**

1. Does the manuscript provide a valid rationale for the proposed study, with clearly identified and justified research questions?

Reviewer #3: Yes

2. Is the protocol technically sound and planned in a manner that will lead to a meaningful outcome and allow testing the stated hypotheses?

Reviewer #3: Yes

3. Is the methodology feasible and described in sufficient detail to allow the work to be replicable?

Reviewer #3: Yes

4. Have the authors described where all data underlying the findings will be made available when the study is complete?

Reviewer #3: No

5. Is the manuscript presented in an intelligible fashion and written in standard English?

Reviewer #3: Yes

You may also provide optional suggestions and comments to authors that they might find helpful in planning their study.

Reviewer #3: In the abstract of this revision, it states that “5 minutes (T2) after the start of the intervention, and 5 minutes (T3) after the start of the intervention”, which means that both T2 and T3 indicate the same time point as 5 minutes after the start of the intervention. Authors need to carefully check if the label of time point is correct instead of mismatch through the text.

**Do you want your identity to be public for this peer review?** For information about this choice, including consent withdrawal, please see our Privacy Policy

Reviewer #3: No

---

## [Editor Report · Acceptance letter]

PONE-D-25-25104R4

PLOS ONE

Dear Dr. Tang,

I'm pleased to inform you that your manuscript has been deemed suitable for publication in PLOS ONE. Congratulations! Your manuscript is now being handed over to our production team.

Kind regards,

on behalf of

Dr. JONATHAN BAYUO

Academic Editor

PLOS ONE